# Subthalamic Nucleus Deep Brain Stimulation Treats Parkinson’s Disease Patients with Cardiovascular Disease Comorbidity: A Retrospective Study of a Single Center Experience

**DOI:** 10.3390/brainsci13010070

**Published:** 2022-12-29

**Authors:** Changming Zhang, Jiakun Xu, Bin Wu, Yuting Ling, Qianqian Guo, Simin Wang, Lige Liu, Nan Jiang, Ling Chen, Jinlong Liu

**Affiliations:** 1Department of Neurosurgery, The First Affiliated Hospital of Sun Yat-sen University, Guangzhou 510080, China; 2Department of Neurology, The First Affiliated Hospital of Sun Yat-sen University, Guangzhou 510080, China; 3Department of Anesthesiology, The First Affiliated Hospital of Sun Yat-sen University, Guangzhou 510080, China

**Keywords:** Parkinson’s disease (PD), subthalamic nucleus (STN), deep brain stimulation (DBS), cardiovascular disease (CVD), multi-disciplinary team (MDT), general anesthesia

## Abstract

Background: Subthalamic nucleus (STN) deep brain stimulation (DBS) is an effective method for treating Parkinson’s disease (PD). However, safety of STN-DBS treating PD patients with cardiovascular disease (CVD) comorbidity is rarely focused and reported. The aim of this study is to investigate the efficacy and safety of STN-DBS treating PD patients with CVD comorbidity. Methods: We retrospectively included PD patients with CVD comorbidity who underwent STN-DBS under general anesthesia in our center from January 2019 to January 2021. Patient’s PD symptoms and cardiopulmonary function were evaluated by a multi-disciplinary team (MDT) before surgery. Post-operative clinical outcome and complications were collected until 1-year follow-up. Results: A total of 38 patients (26 men/12 women) of mean body mass index (BMI) 24.36 ± 3.11 kg/m^2^, with different CVD comorbidity were finally speculated in the study. These CVD include mainly hypertension, coronary artery disease, thoracic aortic aneurysm, heart valve replacement, pacemaker implantation, atrial fibrillation, patent foramen ovale, and so on. The mean systolic blood pressure (SBP) of 38 patients at admission day, pre-operation day, and discharge day timepoint was 135.63 ± 18.08 mmHg, 137.66 ± 12.26 mmHg, and 126.87 ± 13.36 mmHg, respectively. This showed that blood pressure was controlled well under stable and normal state. The indicators of myocardial infarction Troponin T (Tn T-T) levels at pre-operation, 1 day and 7 days after operation timepoint were 0.014 ± 0.011 ng/mL, 0.015 ± 0.011 ng/mL, and 0.014 ± 0.008 ng/mL, showing no significant fluctuation (F = 0.038, *p* = 0.962). STN-DBS improved PD patients’ UPDRS III scores by 51.38% (t = 12.33, *p* < 0.0001) at 1-year follow-up compared with pre-operative baseline. A total of 11 adverse events were recorded until 1-year follow-up. No obvious cardiovascular complications such as intracranial hematoma or clot-related events occurred within 1 year after surgery except 1 case of hematuria. Conclusions: STN-DBS under general anesthesia is safe and effective for treating PD patients with CVD comorbidity. Our clinical experience and protocol of the MDT offers comprehensive perioperative evaluation for DBS surgery and mitigates bleeding and cardiovascular-associated events in PD patients with CVD comorbidity.

## 1. Introduction

Parkinson’s disease (PD) is a neurodegenerative disease that predominantly affects the elderly [1]. Deep brain stimulation (DBS) of the subthalamic nucleus (STN) has been proved to be an effective surgical method for relieving the symptoms of PD [2,3]. Due to older age and impaired autonomic nerve function, PD patients are often complicated with cardiovascular disease (CVD) [4,5,6]. A report attempted to further explore the links between CVD and Parkinson’s disease, such as older age, male sex, and maybe type 2 diabetes as high risks for both CVD and PD [5]. Another research of 24 h ambulatory blood pressure monitoring also found that 38.13% of PD patients presented high blood pressure [6]. Moreover, research also showed that PD can increase stroke risk significantly [4]. Those reports indicate that there might exist relationship between PD with CVD comorbidity.

As PD patients with CVD comorbidity present not low incidence rate, these patients often take anti-platelet drugs and anticoagulant drugs for reducing the risk of myocardial and cerebral infarction. Management of these drugs in the perioperative period is difficult because they drugs increase the risk of operative bleeding when they are continued or increase the risk of myocardial infarction and stoke when they are discontinued [7,8]. Intracranial hemorrhage is currently considered the most serious complication of DBS [9], which results in prolonged hospital stay and neurological deficits.

Until now, more and more normal PD patients benefit for receiving DBS treatment. In the meantime, PD patients with complicated CVD comorbidity may be encountered frequently. It is known that perioperative risks of death, hemorrhage, myocardial infarction, and stoke of treating those patients are higher than normal PD patients. Very few studies have reported the efficacy and safety of deep brain stimulation in PD patients with CVD comorbidity; more careful clinical experiences of this issue deserve further exploring. Hence, the aim of this study is to investigate efficacy and safety of STN-DBS treating PD patients with CVD comorbidity. Our center already found that the general anesthesia method can improve patients’ comfort and control intra-operative vital signs stably [10]. Hence, we present the hypothesis that (1) the cardiovascular and cerebrovascular function evaluation by a multi-disciplinary team (MDT) before DBS operation can help reduce perioperative accidents. (2) STN-DBS treatment under general anesthesia for PD patients with CVD comorbidity should be safe and effective after careful and precise evaluation.

## 2. Materials and Methods

### 2.1. Participants and Study Design

This is a retrospective observation research. A total of 38 PD patients with different CVD comorbidity who received bilateral STN-DBS under general anesthesia in the First Affiliated Hospital of Sun Yat-sen University from January 2019 to January 2021 was identified from total 113 consecutive DBS cases recruited in this period. An MDT of a neurosurgery doctor, neurology physician, cardiology physician, and anesthesiology physician made careful and complete evaluations for every patient in different specialty aspects during peri-operation period. All 38 cases were turned on around 1 month after DBS surgery and have complete post-operative 1-year follow-up data.

### 2.2. Pre-Operative Cardiovascular Assessment

PD patients taking antiplatelet drugs or anticoagulants were initially evaluated in the outpatient clinic. Those taking anti-platelet drugs were asked to stop taking anti-platelet drugs for 7 days before being admitted to the hospital. Blood pressure monitoring, electrocardiograph (ECG), echocardiography, and blood tests were performed after patients’ admission. Meanwhile, the cardiology physician of the MDT assessed whether the patient’s cardiovascular condition could withstand DBS. In addition, we used the Revised Cardiac Risk Index (RCRI) scale to evaluate the occurrence of perioperative cardiovascular events and the American Society of Anesthesiologists (ASA) classification to evaluate perioperative mortality. Cardiac function was evaluated by electrocardiogram, echocardiography, and myocardial marks such as creatine kinase-myocardial band (CK-MB) with normal range of 0.1~4.94 ng/mL, myo-hemoglobin (MYO) with normal range of 25~75 ng/mL, and troponin T (Tn T-T) with normal range of 0~0.014 ng/mL. Bleeding and coagulation indexes such as prothrombin time (PT) with normal range of 11~14 s (s), activated partial thromboplastin time (APTT) with normal range of 25 ~ 31.2 s, and international normalized ratio (INR) with normal range of 0.8~1.15 were applied for monitoring bleeding risk.

For those who took anticoagulant medication, patients stopped taking anticoagulant drugs and were bridged to low molecular weight heparin after admission when INR was less than 2. Oral anticoagulant drugs were added on the third day after DBS operation. Low molecular weight heparin was withdrawn when INR reached the standard (INR > 1.2). Patients who discontinued anti-platelet drugs for at least one week before surgery started taking them on the third day after DBS surgery.

### 2.3. STN-DBS Procedure

Pre-operative planning and surgical procedures were described in a previous study [10]. All patients underwent 3.0 T magnetic resonance imaging (MRI) scans to identify STN before surgery. On the day of the surgery, a head-computed tomography (CT) scan with a Leksell frame mounted on the skull was performed. The stereotactic CT images were merged with the pre-operative MRI image to plan the surgical coordinates and trajectories using Stereotactic Planning Software (iPlan, Brainlab, Feldkirchen, Germany). The operation was performed under general anesthesia. Microelectrode recording (MER) of bilateral STN were performed during operation. After ensuring the best trajectory, the leads were implanted. Finally, an implantable pulse generator (IPG) was implanted in right-side sub-clavicular area under general anesthesia. Head CT was performed on the morning of the first post-operative day to evaluate whether there was intracranial hemorrhage and pneumocephalus. Head 1.5T-MRI was performed on the first post-operative week to confirm the position of electrode around 1 month after operation when the micro-lesion effect disappeared, the device was switched on and programmed.

### 2.4. Data Collection

The PD symptoms were evaluated by neurology and neurosurgery at 1 week before surgery, 1 month (when the device was turned on) after surgery, and 12 months after surgery. One month after operation, the device was turned on continuously, and each evaluation was performed in the state of “medication-off and stimulation-off” and “medication-off and stimulation-on”. “Medication-off” requires the patient stop taking the anti-PD drug for at least 12 h. Unified Parkinson’s Disease Rating Scale Part 3 (UPDRS III) motor score was used to evaluate the improvement of motor symptoms. The severity of the disease was evaluated by the Hoehn and Yahr (H and Y) scale. Meanwhile, perioperative cardiovascular events and other side events were recorded.

### 2.5. Statistical Analysis

The GraphPad (GraphPad Software, IBM Corp, Armonk, NY, USA, version 9.0) software was used for analysis. We used descriptive analysis to summarize the general data of PD patients. The quantitative data were expressed as mean ± standard deviation (SD), and the qualitative data were expressed as number of cases and percentages. The T test was used to compare the improvement rate of symptoms. A one-way ANOVA analysis was used to compare those blood pressure, myocardial marks, and coagulation indexes at three different timepoints. The Shapiro–Wilk tests were used for passing the normality test (alpha = 0.05) and the Greenhouse–Geisser correction was used to adjust. The Fish’s least significant difference (LSD) test was used for the multiple comparison of data at three different timepoints. Significance can be recognized when *p* < 0.05.

## 3. Results

### 3.1. General Characteristics

From January 2019 to January 2021, a total of 113 patients received STN-DBS in our center, of which 38 patients had cardiovascular diseases, accounting for 33.63%. Baseline characteristics of the 38 cases are listed in Table 1. There were 26 (68.42%) men and 12 women (31.58%). The average age was 65.03 ± 6.89 years old. The average height was 163.2 ± 9.13 cm and the average weight was 65.1 ± 11.12 kg. The average body mass index (BMI) was 24.36 ± 3.11 kg/m^2^, suggesting overweight for the Asian ethnic group when BMI ≥ 24 kg/m^2^. The “medication-off” state average HY stage was 2.7 ± 0.3. The average levodopa equivalent daily doses (LEDDs) were 849.3 ± 235.7 mg. The average duration of PD was 10.03 ± 4.11 years. The average ASA classification was 2.97 ± 0.16. The average Revised Cardiac Risk Index (RCRI) was 0.76 ± 0.82. Moreover, those 38 patients also had other comorbidities, including 6 diabetes mellitus, 5 fractures, 2 cerebral infarction, and 1 gout.

The types of cardiovascular diseases of 38 cases were shown in Table 2. Among them, 24 patients had hypertension only and took a variety of antihypertensive drugs to control blood pressure. Four patients had coronary artery disease (CAD) with stent implantation and were taking aspirin or clopidogrel, and 3 patients had CAD without stent implantation and were taking clopidogrel. All those CAD patients received coronary artery CT angiography to exclude new or severe coronary stenosis before surgery. One patient had dilated heart disease and was taking amlodipine. One patient had deep vein thrombosis (DVT) of both lower extremities and was taking aspirin regularly. One patient had a thoracic aortic aneurysm of 25 mm × 22 mm × 35 mm in pre-operative screening. This patient received stenting surgery first and received DBS in 2 months later. One patient had heart valve replacement and was taking warfarin. One had a pacemaker implantation which is not compatible with MRI. One had a 20-year history of atrial fibrillation with taking aspirin regularly. One patient combined with cerebral infarction was screened to have a patent foramen ovale occasionally and was regularly taking clopidogrel.

The results of the electrocardiogram and echocardiography of those patients are shown in Appendix A. The echocardiograms of all patients were within reasonable limits. The mean left ventricular posterior wall thickness (LVPWT) was 9.16 ± 0.83 mm, which lies in normal range of <11 mm. The mean ejection fraction (EF%) was 69.59 ± 4.96%, which lies in normal range of >50%. Twenty cases of these patients had completely normal ECGs. The remaining had minor abnormalities in the ECG, such as abnormal Q waves, sinus tachycardia with mild ST changes, incomplete right bundle branch block, and left anterior fascicular block (Appendix A), which did not require extra special treatment indicating by cardiology physician. The mean heart rate was 78.53 ± 12.41 beats per minute (bpm), and the mean R-wave in lead V5 (RV5) + S-wave in lead V1 (SV1) was 2.51 ± 0.57 mV, which lies in the normal range of <3.5 mV. This indicates no left ventricular hypertrophy exists. The cardiology physician and anesthesiology physician of the MDT assess that patient’s cardiovascular condition can withstand DBS surgery.

### 3.2. Perioperative Trends in Cardiovascular Measures

#### 3.2.1. Trends in Blood Pressure

The mean systolic blood pressure (SBP) of 38 patients was 135.63 ± 18.08 mmHg on admission day, and 137.66 ± 12.26 mmHg and 126.87 ± 13.36 mmHg before operation and on discharge day, respectively. SBP was within the normal range in all three periods. The mean diastolic blood pressure (DBP) was 78.87 ± 12.53 mmHg on admission day, 81.82 ± 8.06 mmHg before operation, and 73.37 ± 10.01 mmHg on discharge day. Diastolic blood pressure was also within the normal range in all three time periods (Figure 1). Patients also took antihypertensive drugs in surgery day morning. Regular and timely medication control helped keep the blood pressure stable. Intra-operative blood pressure and other vital signs were also controlled under general anesthesia (detailed data not shown).

#### 3.2.2. Trends in Cardiac Markers

The mean CK-MB levels of pre-operation, 1 day and 7 days after surgery were 2.00 ± 0.97 ng/mL, 2.09 ± 1.02 ng/mL, and 1.88 ± 0.99 ng/mL (Figure 2), respectively, showing no difference (F = 0.354, *p* = 0.702). The mean MYO levels of were 40.13 ± 28.30 ng/mL, 92.03 ± 53.33 ng/mL, and 76.70 ± 82.95 ng/mL (Figure 2) at pre-operation, 1 day, and 7 days after operation timepoints, respectively. The post-operative MYO increased significantly higher at first day after operation but quickly went down to near normal level than pre-operation baseline (F = 5.876, *p* < 0.005). The mean Tn T-T levels of pre-operation, 1 day and 7 days after surgery were 0.014 ± 0.011 ng/mL, 0.015 ± 0.011 ng/mL and 0.014 ± 0.008 ng/mL (Figure 2). Three Tn T-T levels lies almost in the normal range with no significant difference (F = 0.038, *p* = 0.962), suggesting no significant fluctuation, and myocardial infarction accidents during the perioperative period is greatly reduced.

#### 3.2.3. Blood Coagulation Indicators

Five of the 38 patients were on long-term anticoagulant or antiplatelet therapy. Those patients received low molecular weight heparin after admission. Monitoring the blood coagulation index is necessary. The mean PT was 13.8 ± 6.10 s, 13.04 ± 0.78 s, and 13.16 ± 1.21 s on admission day, operation day, and 3 days after operation (Figure 3), respectively, showing no significant difference (F = 0.063, *p* = 0.938). The mean APPT was 29 ± 5.18 s, 30.40 ± 5.96 s, and 30.62 ± 3.15 s on admission day, operation day, and 3 days after operation (Figure 3), respectively, showing no significant difference (F = 0.160, *p* = 0.853). The mean INR of admission day was 1.21 ± 0.58, which was higher than normal range of 0.8~1.15 due to anticoagulant or antiplatelet therapy. For reducing bleeding risk, the mean INR decreased to 1.04 ± 0.04 and 1.15 ± 0.15 at operation day and 3 days after operation, respectively. Next, anti-coagulant or antiplatelet therapy was re-started after 3 days of DBS surgery for reducing the thromboembolism event risk. Finally, low molecular weight heparin was withdrawn when INR reached the standard (INR > 1.2). The careful monitoring aims to achieve balance between hemorrhage risks with anticoagulants and thromboembolism with discontinuation.

### 3.3. Clinical Outcome and Post-Operative Complications

UPDRS III score of those patients’ “medication-off and stimulation-on” state after STN-DBS at one year follow-up was 22.87 ± 5.87 (t = 12.33, *p* < 0.0001), obviously declining from baseline score 48.24 ± 11.24 at “medication-off and stimulation-off” state (detail score data not shown). STN-DBS significantly improved patients motor symptom by 51.38 ± 12.39% at one year follow-up.

A total of 11 adverse events were recorded until 1-year follow-up. None of them had cardiovascular-related complications such as intracranial hemorrhage or thromboembolic events. One case of a 74-year-old man with prostatic hyperplasia presented hematuria at one week after DBS surgery, which was most likely caused by multiple urinal catheter due to urinary retention. Two patients had pneumonia 1 month after operation. One patient presented wound infection and exudation 1 month after operation. Poor wound healing occurred in 2 cases. One patient had a seizure 2 weeks after operation. Post-operative pleural effusion with atelectasis occurred in 1 case. Post-operative apathy and urinary incontinence occurred in 1 case. One patient developed post-operative delirium. A patient with an implanted pacemaker presented dopa drug withdrawal syndrome 2 weeks after DBS surgery, thus the DBS was turned on 2 weeks ahead of schedule and dopa drug dose increased temporarily (Table 3).

## 4. Discussion

PD patients with cardiovascular disease (CVD) comorbidity present relative high risk of received DBS surgery. The safety and efficacy of STN-DBS for Parkinson’s disease patients with cardiovascular disease have rarely been reported previously. Hence, the aim of this study is to investigate efficacy and safety of STN-DBS treating PD patients with CVD comorbidity. In this study, PD patients with CVD comorbidity received complete pre-operative safety evaluation first, and then received STN-DBS treatment under electrophysiological monitoring with general anesthesia. All patients achieved expected improvement for PD motor symptoms. No cardiovascular-related complications occurred during the perioperative period of STN-DBS. This manuscript brings new clinical protocol and experience for treating those complicated PD patients with CVD comorbidity.

Previous reports on the management of PD patients with cardiovascular disease mainly focused on the post-admission bridging of anticoagulation and anti-platelet drugs [11,12,13]. However, little attention has been paid to other cardiovascular diseases and pos-operative efficacy. The higher BMI in this PD group is a high risk factor for cardiovascular disease. Surgery-related cardiovascular events mainly include bleeding, ischemia, myocardial infarction, heart failure, etc. Moreover, those patients’ average ASA classification was 2.97 ± 0.16, which indicated perioperative death ratio around 1.82–4.30%. The average Revised Cardiac Risk Index (RCRI) was 0.76 ± 0.82, which indicated the occurrence ratio of perioperative cardiovascular events around 0.4–0.9%. However, no perioperative death and perioperative cardiovascular events happened in this study. We performed EEG, echocardiography, and blood myocardial marks tests after PD patients’ admission. The MDT then fully assessed the neurology system, cardiovascular condition, and pulmonary function. The cardiovascular treatment was adjusted to adapt to the surgery, such as oral anticoagulant drugs were changed to low molecular weight heparin, etc. Previous literature reported that the probability of intracranial hemorrhage of DBS was 1.7~3.4% [9,14,15]. We had no intracranial bleeding events in this study. Interestingly, our recent statistical data of all DBS surgery showed the mean ratio of intracranial hemorrhage was between 0.5% and ~1.0%, though no bleeding event happened in this study. This indicates that the rate of intraoperative and post-operative bleeding events in our center is relative lower than those prior reports [14,15]. The reason can be that blood pressure and blood coagulation indicators are stable during the perioperative cardiovascular evaluation. Furthermore, 3.0 T1 and T2 MRI sequences combined with susceptibility weighted imaging (SWI) were used to double check leads trajectory to avoid vessels in order to reduce bleeding risk. In addition, single-track electrophysiology can reduce the risk of intraoperative bleeding more than multiple-track electrophysiology. However, there were other complications in this group of patients, such as 2 cases post-operative pulmonary infection and 2 cases poor wound healing, which accounted for 5.2% in this study.

Previous studies have shown that DBS is safe and feasible for patients with cardiac pacemaker implantation, and pacemaker implantation should not be considered as a contraindication to DBS [16,17]. In this study, STN-DBS was also performed in one 71-year-old female PD patient with a pacemaker implantation. The patient’s perioperative cardiac function indexes such as blood pressure and heart rate remained stable. The brain stimulator was kept around 20 cm far from cardiac pacemaker implantation in case of mutual interruption. Good improvement of motor symptoms was obtained after DBS turn-on, although drug withdrawal syndrome occurred 2 weeks after surgery.

All 38 patients underwent STN-DBS under general anesthesia. Due to anxiety and sympathetic nerve stimulation, large fluctuations of heart rate and blood pressure might happen in local anesthesia for patients with CVD during surgery. Recent studies have noted that DBS under general anesthesia has no significant effect on the intraoperative electrophysiological signals and post-operative efficacy [18,19]. In this study, the patients were sedated with propofol, analgesia with remifentanil, and sufentanil. The blood pressure and heart rate of 38 patients were well-controlled during DBS surgery period, suggesting that general anesthesia method might also contribute for relative low bleeding risk and offer patients more comfort. STN-DBS has a good effect on relieving tremor and rigidity in PD patients. In this study, DBS improved UPDRS III scores by 51.38 % at 1-year follow-up compared with the pre-operative baseline. Long-term follow-up reports from our center have already shown that patients achieve satisfactory symptom control after STN-DBS treating PD [20,21].

## 5. Limitations and Strengths

There are several limitations for this study. First, this study is a retrospective observational study. Second, the number of this study remains small. Thirty-eight PD with CVD cases were included, with 63.1% combined only with hypertension. Bleeding and thromboembolic risk of those cases with those severe CVD such as coronary artery disease and cardiac rhythm disease should be further focused, offering a safe guideline for treating complicated PD patients. However, this study offers useful clinical experience of STN-DBS for treating PD patients with different CVD comorbidity, with very little prior relative research reported. A larger cohort clinical data are necessary to further explore this current research issue, especially in China with an aging society.

## 6. Conclusions

STN-DBS is safe and effective for treating PD patients with CVD comorbidity, improving PD patients’ motor UPDRS III scores at 1-year follow-up compared with pre-operative baseline. STN-DBS under general anesthesia can help control blood pressure and keep heart rates stable and constant, which offers convenience and comfort for anxious patients or complicated comorbidity PD patients especially with all kinds of CVD. Comprehensive perioperative evaluation of peri-operation death and cardiac incidence risk is necessary for treating PD patients with CVD comorbidity. Our clinical experience and protocol of the MDT offers complete and careful perioperative evaluation for DBS surgery and mitigates bleeding and cardiovascular-associated events in PD patients with CVD, though with a similar non-cardiovascular side event ratio as reported.

## Figures and Tables

**Figure 1 brainsci-13-00070-f001:**
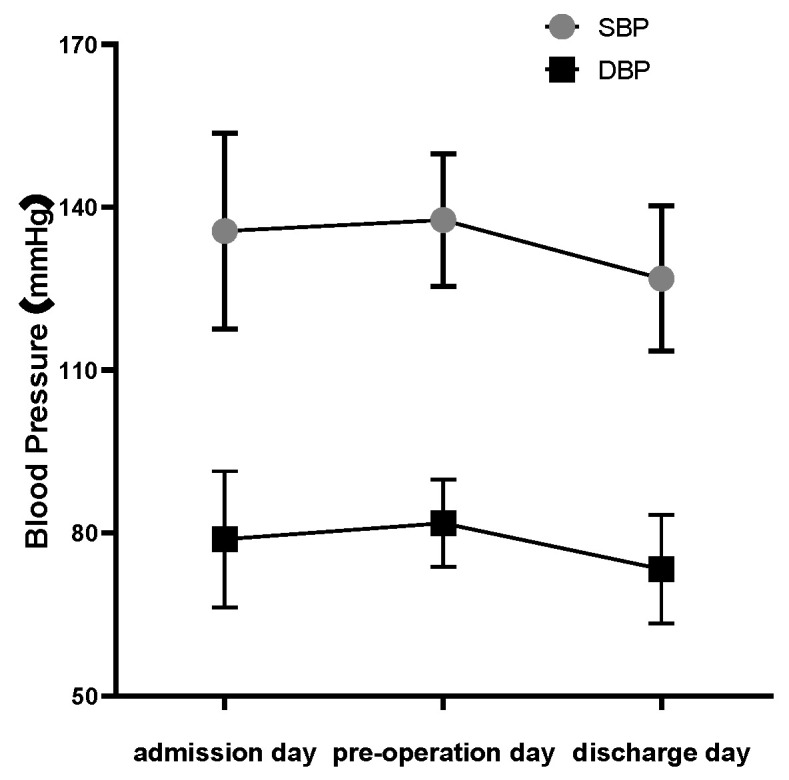
Trends in blood pressure. Systolic blood pressure (SBP) and diastolic blood pressure (DBP) change trends in three different timepoint during the STN-DBS perioperative period. Normal SBP < 140 mmHg; normal DBP < 90 mmHg.

**Figure 2 brainsci-13-00070-f002:**
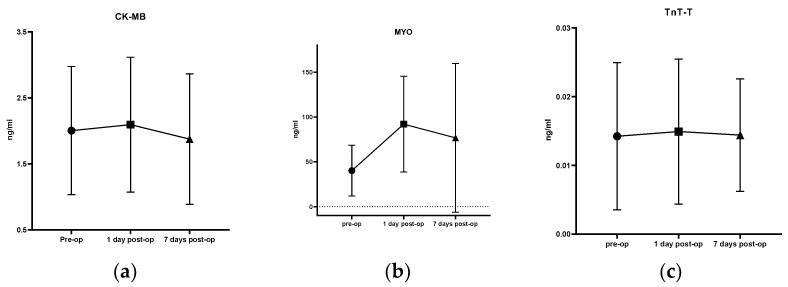
Trends in myocardial markers. This shows trends of the myocardial markers CK-MB (**a**), MYO (**b**), and Tn T-T (**c**) at three different timepoints during the STN-DBS perioperative period. pre-op: pre-operation; post-op: post-operation.

**Figure 3 brainsci-13-00070-f003:**
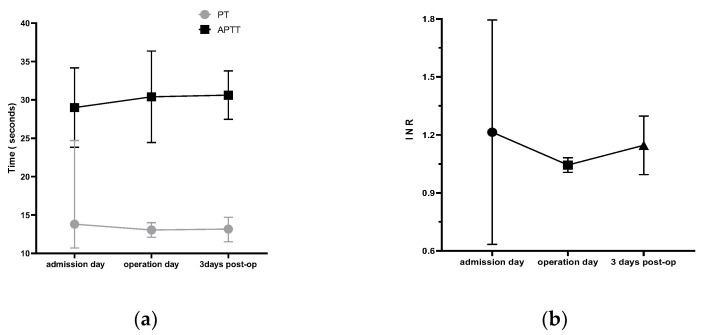
Trends in blood coagulation indicators. It shows trends of the PT, APTT (**a**), and INR (**b**) at three different timepoints during the STN-DBS perioperative period. post-op: post-operation.

**Table 1 brainsci-13-00070-t001:** Population characteristics.

Basic Information	Mean Value	Range
Age (year)	65.03 ± 6.89	49–78
Height (cm)	163.2 ± 9.13	140–181
Weight (kg)	65.1 ± 11.12	40–85.5
BMI (kg/m^2^)	24.36 ± 3.11	16.02–31.83
HY stage	2.7 ± 0.3	2.5–4.0
LEDDs (mg)	849.3 ± 235.7	500–1548
PD course (year)	10.03 ± 4.11	4–22
ASA level	2.97 ± 0.16	2–3
RCRI index	0.76 ± 0.82	1–3
Other comorbidities	6 diabetes mellitus, 5 fractures, 2 cerebral infarction, 1 gout	

BMI: body mass index; HY: Hoehn and Yahr; LEDDs: levodopa equivalent daily doses; PD: Parkinson’s disease; ASA: American Society of Anesthesiologists; RCRI: Revised Cardiac Risk Index.

**Table 2 brainsci-13-00070-t002:** Types of cardiovascular diseases.

CVD Types	Number	CVD Course (Years)	Drugs
Hypertension only	24	8.15 ± 4.89	Antihypertensive medications
CAD (stent implantation) and Hypertension	4	8.75 ± 2.8	Aspirin or clopidogrel
CAD (without stents) and Hypertension	3	8.3 ± 2.5	Clopidogrel
Dilated cardiomyopathy and Hypertension	1	3	Amlodipine
Lower limbs DVT and Hypertension	1	10	Aspirin
Thoracic aortic aneurysm	1	2 months	None
Cardiac valve replacement	1	8	Warfarin
Pacemaker implanted	1	8	None
Atrial fibrillation	1	20	Aspirin
Patent foramen ovale	1	50	Clopidogrel

CVD: cardiovascular disease; CAD: coronary artery disease; DVT: deep vein thrombosis.

**Table 3 brainsci-13-00070-t003:** Post-operative complications.

Cases	Post-Operative Complications	Post-Operative Occur Time	Ratio
2	Pneumonia	1 month	5.3%
1	Wound infection and exudation	1 month	2.6%
2	Poor wound healing	2 weeks	5.3%
1	Seizure	2 weeks	2.6%
1	Hematuria	1 week	2.6%
1	Pleural effusion with atelectasis	1 month	2.6%
1	Apathy and urinary incontinence	1 week	2.6%
1	Delirium	1 week	2.6%
1	Dopa drug withdrawal syndrome	2 weeks	2.6%

## Data Availability

The research data are available upon request.

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
