# Peer review of "Subthalamic Nucleus Deep Brain Stimulation Treats Parkinson’s Disease Patients with Cardiovascular Disease Comorbidity: A Retrospective Study of a Single Center Experience"

_brainsci, 2022, doi:10.3390/brainsci13010070_

Round 1

Reviewer 1 Report

1. The article type should be mentioned in the title.

2. Abstract.

a. Please provide a clear description of the aim of the study.

3. Introduction

a. The safety of STN DBS was already reported in the literature. What does this manuscript bring new to the current literature?

b. It is advised to provide a more explicit description of the aim of the study.

4. Methods

a. Was any specific scale performed to evaluate cardiovascular risk factors related to surgery?

b. Please, reference a study that already performed this methodology “medication-off & stimulation-off" (short for "off state") and "medication-off & stimulation-on" (short for "on state").”

a. Provide a complete description of the software used for statistics. City?

b. How was calculated the power of the study?

c. How were confounding variables evaluated?

d. Describe the distribution of the variables.

5. Results

a. Describe the abbreviations used in the tables as a footnote.

6. Others

a. Why has not performed any correlation?

Reviewer 2 Report

This is a very interesting study investigating the efficacy and safety of subthalamic nucleus deep brain stimulation in Parkinson's disease combined with cardiovascular disease. The paper is well-written and of interest for the journal. However, several minor changes are recommended before considering it for publication. 

Abstract.

1- In the fifth line the authors report that they included PD patients combined with CVD. I prefer to rename it as PD patients with CVD comorbidity. 

2- Were the patients consecutively recruited? Which kind of recruitment did the authors?

3- The vast majority of patients were men. Did the variable gender influence the main results?

Introduction

1- The introduction section is really brief. I would recommend to expand the first paragraph by including some more details about other medical comorbidities. 

2- Why did the authors evaluated results particularly in PD with cardiovascular disease? This should be further explained in the introduction.

3- The main objectives should be described in more detail in the introduction section. The main and secondary hypotheses are also welcome.

Methods

1- The first subsection should be renamed as "Participants and study design". How were the patients recruited? Consecutively or by any criteria of severity?

Results

1- Hypertension is one of the most important cardiovascular risk factors and is the most frequent in the reported sample. Did the authors find any difference in outcome results according to the presence or absence of hypertension? These would be interesting. In other words, can the authors compare patients with and without hypertension?

Conclusions

1- The conclusions section are really brief. Please, expand it and build a new subsection in the discussion called "limitations and strenghts".

Round 2

Reviewer 1 Report

1. It is advised to revise the title. The modification was done in the stable structure of the manuscript.

2. I believe that the authors' replies should be included in the discussion

“While PD patients with cardiovascular disease(CVD)comorbidity present relative high risk for received DBS surgery. How to evaluated risk and perform STN-DBS among those PD patients with CVD comorbidity is rarely focused and reported. This manuscript bring new clinical protocol and experience for treating those complicated PD patients with CVD comorbidity.”

3. The replied manuscript needs to describe the aim clearly. It is advised to personalize one as in the abstract.

Author Response

Please see the attachment,Thank you!
